# A comparative analysis of academic outcomes in blended versus traditional instructional approaches: An examination within the context of the National Medical Licensing Examination

Siyu Zhang[1], Peijin Wang[1], Chang Cui[1], Jinwen Sima[1], Yuanjie Chen[1], Xiaofei Li[1], Bing Zhang[1], Shijia Lu[1], Yulu Zhang[2], Yinping Sun[1], Baosheng Yang[1]*

**1** North Henan Medical University, Xinxiang, China, **2** Henan Normal University, Xinxiang, China

* 12517012@sqmc.edu.cn

## Abstract

### Objective

This research examines the distinctions between the blended teaching approach and the traditional instructional method within the context of pathophysiology, utilizing the Practicing Physician Qualification Examination as a foundational framework. The objective is to elucidate the effects of the blended teaching model and to provide insights and recommendations from diverse perspectives to guide future reforms in medical education.

### Method

The research methodology comprised a comparative analysis of students' academic performance within the institution, supplemented by the administration of a questionnaire survey. The experimental group participated in a blended classroom teaching model designed to align with the content of the Practicing Physician Examination, while the control group adhered to a conventional teaching model based on the identical examination content.

### Result

The students in the blended learning group demonstrated significantly higher performance on both the mid-term and final examinations compared to those in the traditional lecture group. The instructional method showed a substantial impact on student achievement. Additionally, a majority of students in the experimental group perceived that the blended classroom teaching model, designed around the Practicing Physician Examination, enhanced their competence for the examination.

**Data availability statement:** Some of the data in the article are derived from our actual research data. We will present the original questionnaire, midterm scores, final scores, SD and average scores in the supplementary materials.

**Funding:** This study was supported by 1. University-Level Teaching Reform Initiative at North Henan Medical University: An Empirical Investigation into the Blended Teaching Approach for Pathophysiology with a Focus on the Medical Licensing Examination (2023XJJG35). 2. "One Hospital, One Brand" Initiative by the Basic Medical College at North Henan Medical University: (yyyp2024001). 3. Henan Provincial Social Sciences Association: A Study on Developing High-Quality Faculty Teams in Medical Colleges Inspired by the Educators' Ethos (SKL-2024-805). 4. Outstanding Undergraduate Research Project Fund (2024002ZK, 2024004ZK, 2024008ZK, 2024031SK,2026045ZK). 5. Fund for the Cultivation of Outstanding Young Teachers (SQ2022YQJH02)6.Henan Province University Student Innovation Training Project (S202513505014).7.Henan Province Science and Technology Innovation Project (262102310148).

**Competing interests:** The authors have declared that no competing interests exist.

## Conclusion

The adoption of a blended classroom teaching model, focused on the Practicing Physician Examination, has significantly enhanced educational outcomes in pathophysiology and substantially improved students' preparedness for the Practicing Physician Examination.

## Introduction

National health is fundamental to individual well-being and is intricately connected to the attainment of comprehensive modernization. Across all nations, the advancement and improvement of healthcare systems are of utmost importance, with the availability of qualified medical personnel being the most crucial element. Before entering professional practice, physicians are required to complete extensive education and internship training to continuously enhance their knowledge and skills [1]. Nevertheless, the question remains: how can one evaluate a physician's ability to practice independently? The Medical Practitioner Qualification Examination was instituted as a standard to determine whether an individual possesses the necessary competencies to independently provide medical and healthcare services [2].

The National Medical Licensing Examination [3] is an internationally recognized professional entry assessment employed by numerous countries. It functions as an objective evaluation of the competencies of candidates specializing in clinical medicine, assessing their preparedness to undertake the responsibilities of a physician. Moreover, it represents a vital legal requirement for medical practitioners aspiring to secure employment within healthcare institutions. In China, the National Medical Licensing Examination was introduced in 1998, with the first clinical examination administered in September 1999 [4]. This study evaluated candidates' abilities in clinical reasoning, physical examination, and essential clinical procedural skills. The Medical Licensing Examination has been subject to numerous modifications over time [5]. In April 2015, the National Medical Examination Center announced a restructuring of the National Medical Licensing Examination, transitioning from a single assessment to a two-stage examination process. The first stage is scheduled to take place at the end of the senior year for students majoring in clinical medicine, while the second stage is conducted post-graduation, following the completion of one year of residency training in a hospital setting. Eligibility for the second stage is contingent upon successful completion of the first stage, emphasizing the necessity of passing the initial examination [6]. In the theoretical examination of the first stage, questions related to basic medical sciences comprise approximately 40% to 45% of the content. Pathophysiology, as a fundamental component of basic medical sciences, serves as a critical "link course" bridging basic and clinical medicine, thereby holding significant importance in the qualification examination [7–9].

Pathophysiological research primarily investigates the physiological changes that occur in patients during the onset and progression of diseases. This inherently interdisciplinary field integrates multiple foundational disciplines, thereby playing a crucial

role in reinforcing and expanding the core knowledge of basic medical sciences while simultaneously establishing a robust foundation for clinical medicine. Consequently, the study of pathophysiology is of paramount importance in the undergraduate education of medical students. It is the responsibility of educators to ensure that students achieve a comprehensive understanding of the material presented in pathophysiology courses. Currently, there is a consensus among educators and pathophysiology instructors on the necessity for comprehensive and in-depth reforms in pathophysiology education and teaching [10,11]. These reforms aim to align pathophysiology instruction with the national phased examination for practicing physicians.

In recent years, advancements in information technology have rendered mobile devices, such as smartphones and tablets, essential in everyday life and have increasingly influenced educational practices. Blended learning has emerged as a predominant instructional model in higher education globally [12]. In China, blended teaching has transitioned from an emergency response during the pandemic to a strategically central pedagogy actively embraced by universities in the post-pandemic era [13]. Its implementation across the Chinese educational system has expanded rapidly. Numerous mobile teaching platforms have been developed to facilitate this transition, among which Rain Classroom stands out as an intelligent teaching tool designed for this purpose. By integrating mobile technology into classroom settings, Rain Classroom transforms students from passive recipients to active participants, thereby enhancing learning outcomes.

Blended teaching represents an innovative pedagogical model that integrates conventional face-to-face instructional methods with online learning support [14]. This approach synthesizes in-person and digital environments, effectively balancing collective instruction with opportunities for self-directed study. It underscores the pivotal role of educators in guiding, inspiring, and monitoring the educational process while simultaneously emphasizing the initiative, enthusiasm, and creativity of students as active participants in their learning journey. By fostering a learner-centered community and establishing a diversified, practical, and progressive pathway for competency development, blended teaching addresses the limitations inherent in exclusively face-to-face instruction and signifies a crucial direction for the cultivation of high-quality talent [15].

Prior systematic reviews (e.g., Vallée A et al., 2020; Li M et al., 2025) have consistently demonstrated that blended learning yields superior outcomes in knowledge acquisition compared to traditional learning methods in health education [16,17]. However, there is a notable paucity of research concerning pathophysiological blended teaching tailored to medical licensing examinations, as well as a deficiency in studies exploring blended teaching approaches aligned with career objectives. This study employs the content of the Practicing Physician Qualification Examination as the foundational context for instruction and implements a blended teaching approach that integrates both online and offline modalities. By utilizing resources available on the online platform, the study examines the effectiveness of blended teaching in comparison to traditional classroom methods [18,19]. The primary objectives are to enhance students' adaptability to the segmented examination format for practicing physicians, improve the quality of pathophysiology instruction, and promote the comprehensive development of students. Furthermore, this research seeks to provide valuable insights and guidance for teaching reform in higher education institutions, thereby contributing to the advancement of educational reform initiatives.

## 1. Research methods

### 1.1 Research object

This study involved participants from the 2022 undergraduate students in Clinical Medicine and Imaging at North Henan Medical University. Students were selected from classes 1–22 and 117–122, resulting in a total sample size of 876. Within this sample, classes 1–8, 9–16, 17–19, and 117–122 each formed a large teaching group, with a consistent teacher-to-student ratio maintained across these groups. The study employed a quasi-experimental, non-randomized, comparative design. To minimize experimental error, the first four classes within each large teaching group were designated as the experimental group, while the last four classes served as the control group. This selection was informed by the uniform distribution of admission scores across the smaller classes within each large teaching group.

In this study, the experimental groups consisted of Clinical Classes 1–4, 9–12, 117–119, and Imaging Classes 17–19, which implemented a hybrid teaching approach integrating both online and offline methods. This approach was specifically designed to address the pathophysiology examination components of the licensed Physician examination. The primary online learning platform employed was Xuetangyun, with Youmuke serving as a supplementary resource, alongside traditional teaching methodologies. In contrast, the control groups, comprising Clinical Classes 5–8, 13–16, 120–122, and Imaging Classes 20–22, primarily utilized conventional teaching methods, with instructor-led lectures as the main instructional modality. A comparative analysis was conducted between the experimental and control groups concerning class hours, course instructors, and the content and grading of mid-term and final examinations. The analysis revealed no statistically significant differences between the groups (P > 0.05), thereby confirming their comparability.（In China, after each exam, schools require teachers to conduct a grade analysis based on the students' scores. This grade analysis is then shared with the students, and Chinese students are informed.In the Chinese education system, analyzing students' grades to improve teaching quality is a very common practice. Chinese students are informed of this from the very beginning of their schooling.）

## 1.2 Teaching methods

**1.2.1 Teaching methods of the experimental group.** The experimental cohort adopted a blended instructional approach, integrating both online and offline modalities, specifically tailored to align with the Practicing Physician Qualification Examination outline. Online resources, a pivotal component of blended instruction, are employed pre-class, post-class, and during class sessions. As a result of continuous optimization and enhancement by the teaching and research team, a comprehensive online resource system has been developed (**Table 1**)

In accordance with the educational framework centered on learning outcomes, the teaching and research group conducted a comprehensive revision of the "Pathophysiology" syllabus for the Clinical Medicine and Medical Imaging programs. This revision was guided by the 2024 National Medical Licensing Examination syllabus, serving as a foundational

**Table 1. Classification and quantity of online resources.**

| Content | Chapter | Course-ware | Microles-son | Case | Mind map | Exer-cises | Ideological and political elements | Expand resources |
|---|---|---|---|---|---|---|---|---|
| Introduction | Introduction | 1 | 2 | 2 | 3 | 10 | 1 | 1 |
| General Theory | Introduction to Diseases | 1 | 5 | 2 | 3 | 30 | 1 | 1 |
| Basic pathological process | Disorders of Water and Electrolyte Metabolism | 1 | 8 | 4 | 5 | 120 | 2 | 2 |
| | Acid-Base Balance and Acid-Base Disorder | 1 | 8 | 4 | 5 | 120 | 2 | 2 |
| | Hypoxia | 1 | 6 | 2 | 3 | 60 | 2 | 1 |
| | Fever | 1 | 7 | 2 | 3 | 40 | 1 | 1 |
| | Stress | 1 | 7 | 2 | 3 | 40 | 2 | 1 |
| | Ischemia Reperfusion Injury | 1 | 5 | 2 | 3 | 40 | 2 | 1 |
| | Shock | 1 | 5 | 3 | 3 | 100 | 2 | 2 |
| | Disorders of Coagulation and Anticoagulation Balance | 1 | 5 | 2 | 3 | 60 | 1 | 2 |
| Pathophysiology of various systems and organs | Cardiac Insufficiency | 1 | 6 | 3 | 5 | 100 | 2 | 2 |
| | Pulmonary Insufficiency | 1 | 6 | 3 | 5 | 100 | 2 | 2 |
| | Liver Insufficiency | 1 | 8 | 3 | 5 | 80 | 2 | 2 |
| | Renal Insufficiency | 1 | 6 | 3 | 5 | 100 | 2 | 2 |
| Total | | 14 | 84 | 37 | 54 | 1000 | 24 | 22 |

reference. The primary outcomes of this revision include the creation of a curriculum library, micro-lessons, a case library, a knowledge graph, and a mind map. Notably, the courseware library has been meticulously curated by the teaching and research group, with a focus on the key topics outlined in the National Medical Licensing Examination, and has been made accessible to students via Xuetangyun. Furthermore, micro-lessons on pathophysiology have been recorded, highlighting essential components of the practicing physician examination. These resources are designed to support students in both their preparatory and review phases, thereby enhancing their understanding of critical concepts and examination topics relevant to the practicing physician qualification examination. Concurrently, a case repository focused on the key examination topics for practicing physicians was developed. Additionally, knowledge graphs and mind maps were constructed to elucidate the critical areas of the practicing physician qualification examination in Pathophysiology, thereby enhancing students' ability to locate and identify pertinent examination topics.

This course is notable for its integration of essential concepts from the National Medical Licensing Examination into its curriculum, with careful incorporation of these concepts into the instructional framework. Under the guidance of the National Medical Licensing Examination syllabus, the Pathophysiology Teaching and Research Group conducted extensive discussions to develop the chapters and content for a blended teaching methodology, which includes both online and offline components. This approach aims to systematically organize fragmented knowledge, thereby improving students' comprehension of the underlying framework and logic of the subject matter.

**1.2.2 Teaching methods of the control group.** The control cohort engaged in a traditional lecture-based teaching approach, specifically aligned with the examination framework of the Practicing Physician Qualification Examination. Instructors utilized PowerPoint presentations to deliver a comprehensive and systematic coverage of the syllabus content. Instruction was delivered in a conventional classroom setting, where students primarily assumed a passive role, focusing on listening to lectures and taking notes. Classroom interaction was predominantly initiated by instructors posing questions, to which students responded collectively or individually, sometimes facilitated by the Rain Classroom platform. The course included practice questions and answers as part of the curriculum. Students primarily relied on assigned textbooks and lecture notes provided by instructors as their main learning resources. The online learning platform was used solely for downloading course materials and did not include any obligatory online activities or discussions.

### 1.3 Process of teaching

**1.3.1 Preparation before class of the experimental group.** Integrate students from the 2022 undergraduate clinical Medicine program, particularly those registered in Imaging Clinical courses 1–4, 9–12, and 117–119, as well as Imaging courses 17–19, into the curriculum offered on the Xue Tang Yun Pathophysiology Resource Platform and the You Mu Ke platform (see Table 1).

Educators upload their preliminary course materials to ensure a high degree of alignment between the instructional objectives and the key components outlined in the 2023 Medical Licensing Examination syllabus. Particular emphasis is placed on newly introduced and revised key points within the syllabus. One week prior to the commencement of classes, educators upload teaching videos, courseware, and case analyses, and facilitate online discussions focused on key knowledge points, accompanied by corresponding practice questions for the forthcoming instructional content. Simultaneously, chapter test questions are made available to students following their independent study. These test questions include both actual and practice questions from the physician licensing examinations over the past 25 years, specifically pertaining to the knowledge points addressed in the disseminated chapters.

**1.3.2 Implementation in class of the experimental group.** The instructor systematically elucidates the critical components of the practicing physician examination, with a particular focus on the challenging and pivotal elements of the curriculum and addressing frequently misunderstood concepts. Additionally, the instructor actively engages in discussions to address student inquiries. For example, the subject of water and electrolyte metabolism disorders is allocated a total of eight instructional hours, divided into four sessions of two hours each. The third session utilizes a blended learning

 

approach, covering topics such as "water intoxication, edema, and normal potassium metabolism," wherein students independently engage with online resources to enhance their understanding. To further improve learning outcomes, the fourth session concentrates on reinforcing the concepts and causes of edema, water intoxication, and normal potassium metabolism, which are relevant to the examination topics for practicing physicians, within an online learning framework. The pathogenesis of edema, identified as a critical and challenging topic, often confuses students and is also a key examination point for practicing physicians. During the class, knowledge graphs and mind maps will be utilized to assist students in organizing their thoughts. Additionally, the protein content of edema fluid, associated with different pathogenic mechanisms, will be analyzed to introduce the distinction between "exudate and transudate." Finally, two questions will be posed through a case study: "1. What are the characteristics of edema in this pediatric patient? 2. Why did this pediatric patient develop edema?" These questions will prompt students to engage in discussion and provide answers, thereby reinforcing the relevant knowledge points.

### 1.3.3 Review after class of the experimental group.

The distribution of relevant, real-world questions from practicing physicians augments students' understanding of the segmented examination process for medical practitioners. Additionally, the People's Health Question Bank is utilized to replicate genuine examination questions and to generate a series of review questions for students to engage with after class, thereby consolidating their acquired knowledge. Faculty members within the teaching and research department first synthesize and distill the essential knowledge domains relevant to the practicing physician examination. During the question formulation process, the selection and compilation of questions are rigorously aligned with these identified knowledge domains. For example, the knowledge point "Massive loss of gastric and intestinal fluid is the main cause of hypokalemia" [20] was featured in the medical licensing examinations of 2003, 2012, and 2013. Educators in the teaching and research office consolidated these instances and used the most recent examination question as an illustrative example: "A 30-year-old male presents with a six-year history of recurrent upper abdominal pain, occurring on an empty stomach and at night. The patient's condition deteriorated, presenting with vomiting and diarrhea persisting for two weeks." The primary electrolyte imbalance likely to be observed in this patient is: A. Hypercalcemia; B. Hypokalemia; C. Hypomagnesemia; D. Hyperkalemia; E. Hypernatremia. This scenario was provided to students as a practice exercise, with an emphasis on the high prevalence of this topic in the medical licensing examination, to improve their preparednesand adaptability for the test S1 Fig.

### 1.3.4 Classroom arrangement for the control group.

**Preparation for Class in the Control Group:** Teachers collaboratively prepare PowerPoint presentations and lesson plans. Additionally, they utilize the Rain Classroom platform to introduce new chapters.

**Implementation in Class of the control group:** The classroom dynamics are predominantly influenced by the teacher's instructional approach. Interaction between students and the teacher primarily consists of the teacher posing questions, to which students respond collectively, or individual students are selected to provide answers.

**Review after Class of the control group:** There will be practice questions in the class, and at the end of the class, there will be a few thinking questions, which are mainly from the textbook and have answers.

## 1.4 Evaluation method

### 1.4.1 Performance assessment.

For both the final and mid-term examinations, identical test papers were administered to the experimental and control groups, ensuring a distinct separation between instruction and assessment. The mid-term examination was meticulously designed to assess students' proficiency in preparing for the National Medical Licensing Examination. The test paper closely mirrored the format of actual questions from the National Medical Licensing Examination, with the types of questions and their respective weightings adjusted to accurately reflect the distribution of instructional hours across each chapter. The mid-term examination, conducted in a closed-book format, lasted 60 minutes and consisted of 80 multiple-choice questions, including 40 AI questions, 10 A2 questions, and 30 B1 questions, collectively totaling 100 points. To enhance the clinical response, problem analysis, and problem-solving skills of medical students during their practical training, the final examination has been restructured to incorporate a diverse array of

question formats. The examination consists of 30 A1 questions, collectively worth 30 points; 20 A2 questions, collectively worth 20 points; 2 short-answer questions, contributing a total of 20 points; and a case analysis question comprising 3 sub-questions, collectively accounting for 30 points. The examination is conducted over a period of 120 minutes, with an aggregate score of 100 points, and is administered in a closed-book format.

**1.4.2 Questionnaire survey.** The initial Questionnaire Survey was generated based on a comprehensive literature review of blended learning evaluations and discussions with subject matter experts (experienced pathophysiology teachers), subsequently utilizing Wenjuanxing for data collection and statistical analysis. The questionnaire facilitated an anonymous survey among students in the experimental cohort to ascertain their subjective assessments of the pathophysiological blended teaching approach, which is aligned with the practicing physician qualification examination. This initiative aimed to enhance comprehension of the instructional context and enable the educators to refine their pedagogical strategies accordingly. The Reliability Analysis results that Cronbach's alpha is greater than 0.7 when the questionnaire is available.

**1.4.3 Statistical processing.** Measurement data were presented as mean ± standard deviation (M ± SD). An independent samples t-test was conducted, and statistical analyses were performed using SPSS version 16. A p-value of less than 0.05 was considered to indicate statistical significance. Multiple statistical comparisons were performed on the primary outcomes (Mid-term exam score, final exam score and overall satisfaction score). To control the increased risk of Type I error (family-wise error rate) associated with multiple comparisons, a Bonferroni correction was applied. The significance level was adjusted to $p < 0.025$ (0.017/ 3)

## 2. Teaching effect

### 2.1 Mid-term examination results

The distribution of mid-term examination scores for both the experimental and control groups is presented in Table 2. The examination consisted of objective questions similar to those found in the Practicing Physician Qualification Examination. In an academic context, the sentence can be revised as follows: The students in the blended learning cohort demonstrated significantly higher performance on the mid-term examination (M = 66, SD = 1.708) compared to those in the traditional lecture cohort (M = 62.7, SD = 2.217), as evidenced by t(876) = 2.501, $p = 0.047$, and Cohen's d = 2.01. The 95% confidence interval [0.218, 3.806] further supports the robustness of this finding, as it does not encompass zero, indicating a stable and significant effect size. The Cohen's d value of approximately 2.01 is notably larger than the conventional threshold for a large effect size (0.8), underscoring the substantial impact of the blended learning approach. These results imply that the integration of blended learning with the Practicing Physician Examination enhances the effectiveness of pathophysiology instruction, leading to improved student performance. Additionally, the adaptability of Grade 22 clinical medicine students to the examination's question formats appears to be superior in the blended learning context. Importantly, this difference retained statistical significance following Bonferroni correction for multiple comparisons (t(876) = 2.95, $p = 0.047$, with a significance threshold of $p < 0.017$).

### 2.2 Final examination results

The distribution results of the final examination scores between the experimental group and the control group are shown in the following table (Table 3). The question types of the final exam are objective questions and subjective questions.

**Table 2. Mid-term examination scores of pathophysiology in the experimental group and the control group.**

| Mid-term performance | Variable | | n | M ± SD | t | p | Cohen's d | 95% CI |
|---|---|---|---|---|---|---|---|---|
| | Teaching methods | Experimental group | 445 | 66 ± 1.708 | 2.501 | 0.047 | 2.01 | 0.218,3.806 |
| | | Control group | 431 | 62.7 ± 2.217 | | | | |

**Table 3. Final examination scores of Pathophysiology in the experimental group and the control group.**

| Final performance | Variable | | n | M±SD | t | p | Cohen's d | 95% CI |
|---|---|---|---|---|---|---|---|---|
| | Teaching methods | Experimental group | 444 | 77±1.155 | 3.667 | 0.011 | 2.894 | 0.821, 4.973 |
| | | Control group | 431 | 74.25±0.96 | | | | |

The objective questions are of the type of the practicing physician qualification examination. Subjective questions include short-answer questions and case analysis questions. Students in the blended learning group scored significantly higher on the final examination (M = 77, SD = 1.155) than those in the traditional lecture group (74.25, SD = 0.96), t(876) = 3.667, p = 0.011, Cohen's d = 2.894, 95% CI [0.821, 4.973].The experimental group achieved a higher average score than the control group, with the difference being statistically significant.The Cohen's d approx 2.894 was considered to be extremely large (much larger than the "large effect" criterion of 0.8), and the 95% confidence interval did not include zero, indicating that the effect strength of the difference between the two groups was stable and significant.The results of the independent sample t-test showed that the scores of the experimental group were higher than those of the control group, and there was statistical significance. Thus, it can be seen that the blended teaching based on the practicing physician examination has improved the teaching effect of pathophysiology, and the adaptability of the Grade 22 clinical medicine students to the question types of the practicing physician examination has been enhanced. In addition, blended teaching make the students of thinking independently and expressing themselves better,and this difference remained statistically significant after Bonferroni correction for three comparisons (t(876) = 2.95, p = 0.011, significance threshold p < 0.017).

## 2.3 Assessment of the blended classroom teaching model based on the practicing physician examination by students in the experimental group

In contrast to conventional pedagogical approaches, blended learning prioritizes student-centered educational experiences. Therefore, the efficacy of blended learning should be predominantly evaluated based on student satisfaction with the course. This satisfaction is largely reflected in their post-class feedback and opinions, their perspectives on the design and resources of online open course platforms, their perceptions and understanding of the blended teaching implementation, and their self-assessment of learning outcomes. Consequently, the course is assessed through the administration of a questionnaire survey.

A total of 400 questionnaires were distributed, resulting in the retrieval of 342 valid responses, yielding a recovery rate of 98% (Table 4).The reliability analysis showed that Cronbach's aerfa was 0.725, the questionnaire was valid, and the results were available. The findings indicate that 79% of the students almost adapted to the blended teaching approach for pathophysiology, which is structured around the content of the Practicing Physician Qualification Examination. Furthermore, the educational resources provided by Xuetangyun are perceived as beneficial by 77% of the students in their study of pathophysiology. Additionally, the integration of the practicing physicians' examination syllabus into the pathophysiology curriculum appears to effectively stimulate students' intellectual curiosity. A majority of students find the syllabus to be clear and explicit in the context of pathophysiology instruction. Moreover, 98.7% of the students believe that studying pathophysiology enhances their understanding of clinical knowledge. The overall student satisfaction rate concerning the teaching attitude, methods, and content delivered by pathophysiology instructors is notably high, at 93.6%.

**Table 4. The result of the reliability analysis of the questionnaire survey.**

| Sample capacity | number | Cronbach's aerfa |
|---|---|---|
| 342 | 6 | 0.725 |

## 3. Result

### 3.1 Blended classroom teaching was associated with students' academic performance

The reconfiguration of the blended classroom approach for teaching pathophysiology, guided by the Practicing Physician Qualification Examination, has successfully deconstructed the conventional didactic and unidimensional instructional model. This pedagogical transformation underscores a shift from a "teacher-centered" to a "student-centered" learning paradigm. Given that medical undergraduates are anticipated to sit for the Practicing Physician Qualification Examination in the future, the format and content of pathophysiology assessments will undergo systematic revision. The mid-term examination will integrate question formats from the National Medical Licensing Examination, while the final examination will include both subjective and objective questions.

The most direct and significant metric for evaluating the effectiveness of teaching reforms is the students' examination performance. Analysis of the mid-term and final examination results in pathophysiology for the 2022 cohort of clinical majors revealed that the experimental group outperformed the control group, with statistically significant differences. These findings preliminarily suggest that the reform of the blended classroom approach in pathophysiology, designed to align with the licensed physician qualification examination, is effective. Furthermore, the variation in question types between the mid-term and final examinations supports the notion that this teaching reform contributes to the development of students' clinical reasoning, problem analysis, and problem-solving skills in a clinical context.

### 3.2 Blended teaching is positively correlated with teachers' teaching abilities

The National Medical Licensing Examination has evolved from a singular, external assessment into a bifurcated evaluation process administered both internally and externally to the academic institution. This transformation imposes heightened standards and expectations on students as well as on the infrastructure of higher medical education. The integration of "blended teaching" offers a distinctive opportunity for educators to refine their professional competencies. It is essential for educators to seize this opportunity, identify methodologies that align with their strengths, and develop these into effective pedagogical strategies to enhance student learning outcomes.

In the context of implementing a blended teaching approach for pathophysiology aligned with the National Medical Licensing Examination, the Pathophysiology Teaching and Research Section has developed a comprehensive pathophysiology knowledge graph for the clinical medicine curriculum utilizing the Xuetang Cloud platform. This knowledge graph is structured into seven hierarchical levels, encompassing 1,336 distinct knowledge points, 182 associated learning materials, 32 ideological and political elements, and 1,154 exercises. The exercises have been systematically categorized according to their difficulty level. In the development of a knowledge graph, the tool not only assists educators in organizing the interrelationships among knowledge points but also clarifies and elucidates the connections within each chapter, thereby reducing the need for teachers to repeatedly consult textbooks during lesson preparation. Furthermore, when students utilize the knowledge graph, they can identify and focus on the knowledge points where they have deficiencies, allowing them to construct their own personalized knowledge networks. Additionally, each knowledge point is supplemented with detailed explanations, practice questions, and relevant ideological and political context from the course. To some extent, the knowledge graph serves as a substitute for teachers in addressing and guiding students' inquiries.

### 3.3 Blended classroom teaching cultivates students' abilities of autonomous learning and lifelong learning

The "Undergraduate Medical Education Standards" [21] underscore the importance of medical students acquiring advanced communication skills, embracing the principles of self-directed and lifelong learning, and developing competencies in teamwork and leadership. The National Medical Licensing Examination [22] assesses not only students' disciplinary knowledge and clinical skills but also integrates various components such as self-directed learning, collaborative learning, and communication skills. The necessity for a holistic evaluation of these competencies highlights the urgent

need for reform in medical education methodologies. A notable advantage of blended learning is its ability to promote autonomous learning, lifelong learning, and independent thinking among undergraduate students. This is supported by data from Wenjuanxing, which reveals that 70.4% of students reported improvements in their self-directed learning abilities, while 28.1% observed enhancements in their communication, expression skills, and independent thinking as a result of blended teaching.

### 3.4  The integration of pathophysiological concepts into a blended classroom teaching approach, specifically designed to align with the National Medical Licensing Examination, is beneficial for students' capacity to adapt to the examination requirements

In comparison to previous iterations, the 2024 National Medical Licensing Examination Syllabus has primarily revised the second section, Basic Medicine Comprehensive. This revision incorporates foundational medical content pertinent to various diseases from the original Clinical Medicine Comprehensive, with an updated emphasis on aligning more closely with clinical applications while preserving a comprehensive knowledge framework. Concurrently, the syllabus underscores the significance of integrating basic and clinical knowledge. Pathophysiology, recognized as a "bridge discipline" between basic and clinical medicine and a "compulsory discipline" in the Medical Practitioner Qualification Examination, is assuming an increasingly pivotal role in the examination process. Pathophysiology is a comprehensive interdisciplinary subject that integrates multiple foundational disciplines, playing a crucial role in consolidating and reinforcing the core concepts of basic medical sciences. For example, in the first unit of the 2012 examination, question 17 addresses the physiological response to excessive sweating, specifically the reduction in urine output. The primary reason for this is option B: an increase in plasma osmotic pressure, which leads to elevated secretion of antidiuretic hormone (ADH). While this topic is traditionally associated with physiology, it is also explored in Chapter 3 of Pathophysiology, which examines the types and mechanisms of dehydration and the pathophysiological basis of edema in water and electrolyte imbalances. By consciously reviewing, summarizing, and integrating these topics during instruction, educators can enhance students' understanding and retention of these foundational concepts. Consequently, the extent to which students master pathophysiology some influences their success in passing the practicing physician examination.

## 4.  Limitation

This study possesses several limitations that warrant consideration when interpreting the results. Primarily, the quasi-experimental design, although practical, does not eliminate the potential impact of unmeasured confounding variables. Despite controlling for admission scores and key demographic factors, other elements, such as subtle variations in instructor enthusiasm and teaching effectiveness between the two groups, or differences in students' prior knowledge and self-directed learning skills not captured by admission scores, may have influenced the outcomes. Additionally, the generalizability of our findings is potentially constrained. The research was conducted at a single medical college and exclusively addressed the subject of pathophysiology. Consequently, the results may not be directly applicable to other institutions with different student populations and educational cultures, or to other medical subjects with distinct pedagogical requirements. Future multi-institutional, multi-disciplinary randomized controlled trials are necessary to validate these findings and enhance their external validity.

## 5.  Discussion and prospects

As contemporary society progresses, the penetration rate of the Internet has markedly increased. This development has led to the gradual integration of the Internet into conventional educational methodologies, thereby prompting innovative pedagogical reforms. The adoption of online tools has significantly improved the efficiency of instruction and the accessibility of student learning. This is consistent with the conclusion of a systematic review of 34 studies by Ashraf MA et al. (2021). It further confirmed that teachers use "blended course(s) with a certain applied framework or model" can benefit

their blended learning teaching, students use "blended course(s) with a certain applied framework or model" can foster learning and academic achievement, promote stronger cognitive engagement and interaction [23]. This development signifies a promising advancement for both educators and learners. For educators, the process of homework assessment has transitioned beyond traditional manual correction methods. By employing software, educators can efficiently evaluate and analyze assignments, providing students with immediate feedback. This technological integration significantly enhances teaching efficiency, granting educators a more comprehensive understanding of students' comprehension levels and areas of difficulty. Furthermore, the capability to monitor homework submissions through backend data reduces the administrative burden on teachers. The decrease in time devoted to grading assignments allows for a greater emphasis on instructional design and pedagogical adjustments. For students, the adoption of blended learning facilitates the review and reinforcement of concepts that have not been fully grasped, thereby improving learning efficiency and mastery of the subject matter.

The primary focus of the blended teaching approach discussed in this article is its foundation on the content of the Practicing Physician Qualification Examination for instructional design. Drawing upon our findings and supported by international literature, such as the systematic review by Li S et al. (2024) involving 44 studies, it is evident that encouraging students to engage in pre-class preparation and post-class review enhances their academic performance. This teaching approach, characterized by its novelty and the provision of clear learning objectives, has been shown to be effective [24]. Building on this, we propose a more detailed perspective: the use of mind maps and knowledge graphs further aids students in intuitively understanding the key elements of examinations. The utilization of mind maps and knowledge graphs facilitates a more intuitive comprehension of the examination's key elements for students. Furthermore, the integration of subsequent case studies and assignments reinforces these critical points, thereby enhancing students' mastery of essential knowledge. This approach ultimately aims to improve students' adaptability to the National Medical Licensing Examination.

This study is subject to several limitations. Primarily, the quasi-experimental design, which lacks randomization, constitutes a significant limitation. Despite our efforts to ensure comparability between the intervention and control groups with respect to baseline characteristics, such as prior academic performance and demographics, the absence of randomization may allow unmeasured confounding variables to affect the outcomes. Future research would be enhanced by employing a randomized controlled trial design to bolster the causal inference concerning the efficacy of the blended teaching approach.

In future developments, the research team aims to incorporate artificial intelligence into educational practices [25,26]. Currently, artificial intelligence is increasingly permeating various sectors, and its application in education is deemed essential. By integrating intelligent medical engineering, the reform of pathophysiology education can be further advanced to achieve improved pedagogical outcomes. A satisfaction survey revealed that some students favor using the Bilibili platform for learning. Consequently, the teaching and research team will analyze the online courses accessed by these students and recommend more appropriate courses to enhance their educational satisfaction.

## Supporting information

**S1 Fig. Instructional Design is S1 Fig Title.** Schematic diagram of the blended teaching process integrating online and offline activities.The diagram illustrates the sequential teaching workflow across three phases: Before Class, In-Class, and After Class. Blue boxes represent teacher activities, while pink boxes represent corresponding student activities. The arrows indicate the progression of time and the interactive logic between teaching and learning steps. (PQE: Physician Qualification Examination) is the S1 Fig legend.
(PDF)

**S1 File. Funding Statement is File1 Title.**
(PDF)

**S2 File. Questionnaire Survey Supplementary Material is File2 Title.**
(PDF)

**S3 File. Example of Pathophysiology Knowledge Graph is File3 Title.**
(PDF)

**S4 File. Research data is File4 Title.** Midterm Scores, Final Scores, and Average Scores are the S4 File legend.
(PDF)

**S5 File. Grading of Test Difficulty Coefficients is File5 Title.**
(PDF)

**S6 File. Original questionnaire is File6 Title.**
(PDF)

## Author contributions

**Conceptualization:** Siyu Zhang, Peijin Wang, Yuanjie Chen, Xiaofei Li, Bing Zhang, Yulu Zhang, Baosheng Yang.

**Data curation:** Siyu Zhang, Peijin Wang, Jinwen Sima, Yuanjie Chen, Xiaofei Li, Bing Zhang, Shijia Lu, Yulu Zhang, Baosheng Yang.

**Formal analysis:** Siyu Zhang, Jinwen Sima, Xiaofei Li, Baosheng Yang.

**Writing – original draft:** Siyu Zhang, Peijin Wang, Chang Cui, Jinwen Sima, Yuanjie Chen, Xiaofei Li, Bing Zhang, Shijia Lu, Yulu Zhang, Yinping Sun, Baosheng Yang.

**Writing – review & editing:** Siyu Zhang, Jinwen Sima, Yuanjie Chen, Xiaofei Li, Yinping Sun, Baosheng Yang.

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
