## [Decision Letter · Decision Letter 0]

22 Oct 2025

Dear Dr. Zhang,

Kindly adhere to the formatting of the journalCheck the language and revise the manuscript as indicated by one of the reviewersRevise the manuscript and comply with the reviewers comments in order to ensure that the manuscript is complying with the ethical standards as well as scientific research standards.

plosone@plos.org. . . . A rebuttal letter that responds to each point raised by the academic editor and reviewer(s). You should upload this letter as a separate file labeled 'Response to Reviewers'.A marked-up copy of your manuscript that highlights changes made to the original version. You should upload this as a separate file labeled 'Revised Manuscript with Track Changes'.An unmarked version of your revised paper without tracked changes. You should upload this as a separate file labeled 'Manuscript'.

We look forward to receiving your revised manuscript.

Kind regards,

Ghilan Al Madhagy Ghilan Taufiq Hail, Ph.D.

Academic Editor

PLOS ONE

Journal Requirements:

“This study was supported by 1. University-Level Teaching Reform Initiative at Yubei Medical College: An Empirical Investigation into the Blended Teaching Approach for Pathophysiology with a Focus on the Medical Licensing Examination (2023XJJG35). 2. "One Hospital, One Brand" Initiative by the Basic Medical College at Yubei Medical College: (yyyp2024001). 3. Henan Provincial Social Sciences Association: A Study on Developing High-Quality Faculty Teams in Medical Colleges Inspired by the Educators' Ethos (SKL-2024-805). 4. Outstanding Undergraduate Research Project Fund (2024002ZK, 2024004ZK, 2024008ZK, 2024031SK). 5. Fund for the Cultivation of Outstanding Young Teachers （SQ2022YQJH02）”

Reviewers' comments:

Reviewer's Responses to Questions

**Comments to the Author**

1. Is the manuscript technically sound, and do the data support the conclusions?

Reviewer #1: Yes

Reviewer #2: Yes

2. Has the statistical analysis been performed appropriately and rigorously?

Reviewer #1: Yes

Reviewer #2: I Don't Know

3. Have the authors made all data underlying the findings in their manuscript fully available?

Reviewer #1: Yes

Reviewer #2: Yes

4. Is the manuscript presented in an intelligible fashion and written in standard English?

Reviewer #1: Yes

Reviewer #2: Yes

Reviewer #1: 1. Provide reference for : In China, the National Medical

Licensing Examination was inaugurated in 1998, with the inaugural clinical

examination conducted in September 1999

2. Authors should place 1. Research object under Research method.

3. Participants were drawn from the 2022 undergraduate cohorts of Clinical

Medicine and Imaging, specifically from classes 1-22 and 117-122, totaling 876

students' - authors may need to mention the participants from only 1 university/college - to provide better clarity for international readers.

4. Authors could provide Section : Literature review - write up on blended learning and traditional learning globally and narrow down to China context

Reviewer #2: The manuscript addresses an important and timely issue in medical education: the effectiveness of blended learning compared to traditional teaching, specifically in preparing students for the National Medical Licensing Examination. The topic is relevant to a global readership, and the study has potential to contribute to the literature on instructional reform in medical education. However, the manuscript requires substantial revisions to meet the methodological and reporting standards expected of PLOS ONE.

Major Comments

1. Study Design and Methodology

• The study is described as a comparative analysis, but it is essentially a quasi‑experimental design without randomization. This limitation should be explicitly acknowledged in the Methods and Discussion sections.

• The allocation of classes to experimental and control groups appears based on convenience (first four vs. last four classes). This introduces potential selection bias. Please clarify how baseline equivalence was ensured beyond admission scores.

• The description of the intervention (blended teaching) is detailed, but the control condition is less thoroughly described. For transparency, provide equal detail on the control group’s teaching methods.

2. Ethics and Approvals

• The manuscript currently states “N/A” for ethics approval. Given that students were surveyed and their academic performance analyzed, institutional ethics approval (or a waiver) is typically required. Please clarify whether approval was obtained, and if not, justify why it was deemed unnecessary.

3. Statistical Analysis

• The statistical methods are limited to independent t‑tests. Given the large sample size (n=876), effect sizes (e.g., Cohen’s d) should be reported to contextualize the practical significance of findings.

• Multiple outcomes (mid‑term, final, questionnaire responses) are analyzed, but no correction for multiple comparisons is mentioned. Please address this.

• The reporting of p‑values is inconsistent (e.g., “0.046 5”). Ensure correct formatting and precision.

4. Results Presentation

• Tables are not formatted according to journal standards. For example, Table 2 is fragmented in the draft. Please revise for clarity and consistency.

• The questionnaire results are summarized descriptively (percentages), but no validated instrument is cited. Was the survey piloted or validated? This should be clarified.

5. Discussion and Interpretation

• The discussion tends to overstate the causal impact of blended learning. Given the quasi‑experimental design, conclusions should be tempered to emphasize association rather than causation.

• The manuscript would benefit from situating findings within the broader international literature on blended learning in medical education (e.g., systematic reviews, meta‑analyses).

• Limitations are acknowledged but should be expanded to include issues of generalizability (single institution, single subject area) and potential confounders (teacher effects, prior student ability).

6. Data Availability

• The current statement (“Some of the data… original questionnaire in supplementary materials”) does not meet PLOS ONE’s open data policy. Authors must provide either a public repository link or a clear justification for restrictions.

Minor Comments

• The English language requires editing for grammar and conciseness. For example, phrases such as “the adaptability of 22nd grade clinical medicine major students” should be revised for clarity.

• The abstract should include specific statistical results (means, SDs, p‑values) rather than general statements.

• Figures (e.g., instructional design diagrams) should be redrawn for clarity and labeled according to journal style.

• References: Ensure consistency in formatting (some Chinese references lack full details in English).

Recommendation

Major Revision

The manuscript addresses a relevant and important topic, but significant revisions are required to strengthen methodological transparency, statistical rigor, and adherence to reporting standards. With these improvements, the study could make a valuable contribution to the literature on blended learning in medical education.

.

Reviewer #1: **Yes:** A. DevisaktiA. DevisaktiA. DevisaktiA. Devisakti

Reviewer #2: No

T**o ensure your figures meet our technical requirements, please review our figure guidelines:**
https://journals.plos.org/plosone/s/figures https://journals.plos.org/plosone/s/figures https://journals.plos.org/plosone/s/figures https://journals.plos.org/plosone/s/figures
https://journals.plos.org/plosone/s/figures https://journals.plos.org/plosone/s/figures https://journals.plos.org/plosone/s/figures https://journals.plos.org/plosone/s/figures

---

## [Author Response · Author response to Decision Letter 1]

19 Jan 2026

Dear Reviewers:

On behalf of my co-authors, we thank you for giving us a chance to revise and improve the quality of our article.

We have read the reviewers' and your comments carefully and have made revision which marked in red in the paper. We have tried our best to revise our manuscript according to the comments: “A Comparative Analysis of Academic Outcomes in Blended versus Traditional Instructional Approaches: An Examination within the Context of the National Medical Licensing Examination. (PONE-D-25-34517)”.

Attached please find the revised version, which we hope that you will find this updated manuscript to your satisfaction and consider it for publication as an article in PLOS ONE. Here is a point-by-point response to the reviewers' comments and concerns.

Thank you for taking the time to consider our research and we look forward to hearing from you at your earliest convenience.

Sincerely,

Siyu Zhang

North Henan Medical University,

Xinxiang, HeNan China

E－mail:shiyu15937135473@163.com

---

## [Decision Letter · Decision Letter 1]

19 Mar 2026

Dear Dr. Zhang,

We look forward to receiving your revised manuscript.

Kind regards,

Mukhtiar Baig, Ph.D.

Academic Editor

PLOS One

Journal Requirements:

Reviewers' comments:

Reviewer's Responses to Questions

**Comments to the Author**

Reviewer #1: All comments have been addressed

Reviewer #2: All comments have been addressed

2. Is the manuscript technically sound, and do the data support the conclusions?

Reviewer #1: Yes

Reviewer #2: Yes

3. Has the statistical analysis been performed appropriately and rigorously?

Reviewer #1: Yes

Reviewer #2: Yes

4. Have the authors made all data underlying the findings in their manuscript fully available?

Reviewer #1: Yes

Reviewer #2: Yes

5. Is the manuscript presented in an intelligible fashion and written in standard English?

Reviewer #1: Yes

Reviewer #2: Yes

Reviewer #1: 1. Authors need to remove data / numbers in abstract section , just provide summary of findings

2. Authors could proofread the manuscript.

Reviewer #2: Accept the paper. It is an important topic for the publication in the plos one and I wish the authors well.

.

Reviewer #1: **Yes:** devidevidevidevi

Reviewer #2: No

---

## [Author Response · Author response to Decision Letter 2]

19 Mar 2026

Response Letter

Manuscript Number:PONE-D-25-34517.

Title:A comparative analysis of academic outcomes in blended versus traditional instructional approaches: An examination within the context of the National Medical Licensing Examination.

Mar. 20th, 2026

Dear Editor Mukhtiar Baig, Ph.D., devi and Reviewers:

On behalf of my co-authors, we thank you for giving us a chance to revise and improve the quality of our article.

We have read the reviewers' and your comments carefully and have made revision which marked in red in the paper. We have tried our best to revise our manuscript according to the comments: “A Comparative Analysis of Academic Outcomes in Blended versus Traditional Instructional Approaches: An Examination within the Context of the National Medical Licensing Examination. (PONE-D-25-34517)”.

Attached please find the revised version, which we hope that you will find this updated manuscript to your satisfaction and consider it for publication as an article in PLOS ONE. Here is a point-by-point response to the reviewers' comments and concerns.

Thank you for taking the time to consider our research and we look forward to hearing from you at your earliest convenience.

Sincerely,

Siyu Zhang

North Henan Medical University,

Xinxiang, HeNan China

E－mail:shiyu15937135473@163.com

1. If the authors have adequately addressed your comments raised in a previous round of review and you feel that this manuscript is now acceptable for publication, you may indicate that here to bypass the “Comments to the Author” section, enter your conflict of interest statement in the “Confidential to Editor” section, and submit your "Accept" recommendation.

Reviewer #1: All comments have been addressed

Response:We are pleased that the reviewer confirms all previous comments have been addressed. Thank you for your thorough and constructive feedback throughout the review process, which has significantly strengthened our manuscript.

Reviewer #2: All comments have been addressed

Response:We appreciate the reviewer's confirmation that all concerns raised in the previous round have been satisfactorily addressed. Your insightful comments have been invaluable in improving our work.

2. Is the manuscript technically sound, and do the data support the conclusions?

Reviewer #1: Yes

Response:We thank the reviewer for confirming that the manuscript is technically sound and that the data support the conclusions. We appreciate this positive assessment of our work.

Reviewer #2: Yes

Response:We are grateful to the reviewer for acknowledging that the experiments were conducted rigorously and that the conclusions are appropriately drawn from the data presented. Thank you for your valuable time and expertise in reviewing our manuscript.

3. Has the statistical analysis been performed appropriately and rigorously?

Reviewer #1: Yes

Response:We thank the reviewer for confirming that the statistical analysis in our manuscript has been performed appropriately and rigorously. We appreciate this positive assessment of our methodological approach.

Reviewer #2: Yes

Response:We are grateful to the reviewer for acknowledging the appropriateness and rigor of our statistical analysis. Your validation of our statistical methods is highly valued.

4. Have the authors made all data underlying the findings in their manuscript fully available?

Reviewer #1: Yes

Response:We thank the reviewer for confirming that all data underlying the findings in our manuscript have been made fully available in accordance with the journal's data policy. We appreciate this verification.

Reviewer #2: Yes

Response:We are grateful to the reviewer for acknowledging that our data availability meets the required standards. As stated in our Data Availability Statement, all relevant data are （e.g., provided in the Supporting Information files with accession number S4_File，S5_File）.

5. Is the manuscript presented in an intelligible fashion and written in standard English?

Reviewer #1: Yes

Response:We thank the reviewer for confirming that our manuscript is presented in an intelligible fashion and written in standard English. We appreciate this positive assessment of the language quality.

Reviewer #2: Yes

Response:We are grateful to the reviewer for acknowledging that the language in our manuscript is clear, correct, and unambiguous. Your validation of the writing quality is highly valued.

6. Review Comments to the Author

Reviewer #1: 1. Authors need to remove data / numbers in abstract section , just provide summary of findings

2.Authors could proofread the manuscript.

Response:We thank the reviewer for the valuable comments and suggestions.

Comment 1: "Authors need to remove data / numbers in abstract section, just provide summary of findings"

Response: Thank you for this suggestion. We would like to clarify a situation. During the first revision, another reviewer suggested adding the data to the abstract. After discussing with the team, we have summarized the data and presented the corresponding conclusions.. The revised abstract now reads:

Objective: This research examines the distinctions between the blended teaching approach and the traditional instructional method within the context of pathophysiology, utilizing the Practicing Physician Qualification Examination as a foundational framework. The objective is to elucidate the effects of the blended teaching model and to provide insights and recommendations from diverse perspectives to guide future reforms in medical education. Method: The research methodology comprised a comparative analysis of students' academic performance within the institution, supplemented by the administration of a questionnaire survey. The experimental group participated in a blended classroom teaching model designed to align with the content of the Practicing Physician Examination, while the control group adhered to a conventional teaching model based on the identical examination content. Result: The students in the blended learning group demonstrated significantly higher performance on both the mid-term and final examinations compared to those in the traditional lecture group. The instructional method showed a substantial impact on student achievement. Additionally, a majority of students in the experimental group perceived that the blended classroom teaching model, designed around the Practicing Physician Examination, enhanced their competence for the examination.Conclusion: The adoption of a blended classroom teaching model, focused on the Practicing Physician Examination, has significantly enhanced educational outcomes in pathophysiology and substantially improved students' preparedness for the Practicing Physician Examination.

Comment 2: "Authors could proofread the manuscript"

Response: Thank you for this suggestion. We have carefully proofread the entire manuscript to correct any typographical or grammatical errors.Eg:The sentence on page 1 , lines 5 “Baosheng Yan” has been revised as follows:Baosheng Yang.

Reviewer #2: Accept the paper. It is an important topic for the publication in the plos one and I wish the authors well.

Response:We sincerely thank the reviewer for the positive assessment and for recommending acceptance of our manuscript. We are pleased that you find our topic important and suitable for publication in PLOS ONE. Your encouraging words are greatly appreciated.

---

## [Editor Report · Decision Letter 2]

25 Mar 2026

A comparative analysis of academic outcomes in blended versus traditional instructional approaches: An examination within the context of the National Medical Licensing Examination

PONE-D-25-34517R2

Dear Dr. Zhang,

We’re pleased to inform you that your manuscript has been judged scientifically suitable for publication and will be formally accepted for publication once it meets all outstanding technical requirements.

Kind regards,

Mukhtiar Baig, Ph.D.

Academic Editor

PLOS One

---

## [Editor Report · Acceptance letter]

PONE-D-25-34517R2

PLOS One

Dear Dr. Zhang,

I'm pleased to inform you that your manuscript has been deemed suitable for publication in PLOS One. Congratulations! Your manuscript is now being handed over to our production team.

Kind regards,

on behalf of

Professor Mukhtiar Baig

Academic Editor

PLOS One